# Impact of Gut Microbiota and SCFAs in the Pathogenesis of PCOS and the Effect of Metformin Therapy

**DOI:** 10.3390/ijms251910636

**Published:** 2024-10-02

**Authors:** Evgenii Kukaev, Ekaterina Kirillova, Alisa Tokareva, Elena Rimskaya, Natalia Starodubtseva, Galina Chernukha, Tatiana Priputnevich, Vladimir Frankevich, Gennady Sukhikh

**Affiliations:** 1V.I. Kulakov National Medical Research Center for Obstetrics, Gynecology and Perinatology, Ministry of Healthcare of Russian Federation, 117997 Moscow, Russia; ed_kirillova@oparina4.ru (E.K.); a_tokareva@oparina4.ru (A.T.); e_rimskaya@oparina4.ru (E.R.); n_starodubtseva@oparina4.ru (N.S.); g_chernukha@oparina4.ru (G.C.); t_priputnevich@oparina4.ru (T.P.); v_frankevich@oparina4.ru (V.F.); g_sukhikh@oparina4.ru (G.S.); 2V.L. Talrose Institute for Energy Problems of Chemical Physics, N.N. Semenov Federal Research Center of Chemical Physics, Russian Academy of Sciences, 119334 Moscow, Russia; 3Lebedev Physical Institute, 119991 Moscow, Russia; 4Moscow Center for Advanced Studies, 123592 Moscow, Russia; 5Laboratory of Translational Medicine, Siberian State Medical University, 634050 Tomsk, Russia

**Keywords:** metabolomics, polycystic ovary syndrome, short-chain fatty acids, pathogenesis, intestinal microbiota, gas chromatography-mass spectrometry

## Abstract

Polycystic ovary syndrome (PCOS) is a complex disorder that impacts both the endocrine and metabolic systems, often resulting in infertility, obesity, insulin resistance, and cardiovascular complications. The aim of this study is to investigate the role of intestinal flora and its metabolites, particularly short-chain fatty acids (SCFAs), in the development of PCOS, and to assess the effects of metformin therapy on these components. SCFA levels in fecal and blood samples from women with PCOS (n=69) and healthy controls (n=18) were analyzed using Gas Chromatography–Mass Spectrometry (GC/MS) for precise measurement. Fecal microbiota were quantitatively detected by real-time polymerase chain reaction (PCR). To assess the efficacy of six months of metformin treatment, changes in the microbiota and SCFAs in the PCOS group (n=69) were also evaluated. The results revealed that women with PCOS exhibited a significant reduction in beneficial bacteria (namely, the *C. leptum* group and *Prevotella* spp.) alongside a notable overgrowth of opportunistic microorganisms (*C. perfringens*, *C. difficile*, *Staphylococcus* spp., and *Streptococcus* spp.). An overproduction of acetic acid (AA, FC=0.47, p<0.05) and valeric acid (VA, FC=0.54, p<0.05) suggests a link between elevated SCFAs and the development of obesity and PCOS. Interestingly, AA in the bloodstream might offer a protective effect against PCOS by ameliorating key symptoms such as high body mass index (r=−0.33, p=0.02), insulin resistance (r=−0.39, p=0.02), and chronic inflammation. Although serum SCFA levels showed non-significant changes following metformin treatment (p>0.05), the normalization of AA in the gut underscores that metformin exerts a more pronounced effect locally within the gastrointestinal tract. Furthermore, the study identified the most effective model for predicting the success of metformin therapy, based on serum concentrations of butyric acid (BA) and VA, achieving a 91% accuracy rate, 100% sensitivity, and 80% specificity. These promising findings highlight the potential for developing targeted interventions and personalized treatments, ultimately improving clinical outcomes for women with PCOS.

## 1. Introduction

Polycystic ovary syndrome (PCOS) is a common endocrine and metabolic disorder characterized by androgen excess, ovulatory dysfunction, and polycystic ovarian morphology [1,2,3,4]. The clinical presentation of PCOS is diverse and can be categorized into four distinct phenotypes [4,5]. The Rotterdam criteria are the recommended standard for diagnosing PCOS [5]. This condition involves a wide range of abnormalities, including insulin resistance (IR), disruptions in gonadotropic and neuropeptide secretion, obesity, type 2 diabetes mellitus, atherogenic dyslipidemia, and an elevated risk of cerebrovascular complications [1,2,6,7]. PCOS is a leading cause of female infertility, and its prevalence has been increasing [7,8]. The rising rates of obesity among PCOS patients further intensify metabolic issues, resulting in disruptions of glucose and lipid metabolism [8].

In recent years, there has been a growing focus on the relationship between microbiota and human health [9]. Numerous studies highlight the role of gut and vaginal microbiota in regulating physiological balance, influencing metabolic protection, maintaining structural integrity, and ensuring histological homeostasis [10,11,12]. The composition of gut microbiota in individuals with PCOS significantly differs from that of healthy women [12]. Disruptions in gut flora are closely linked to various clinical symptoms associated with PCOS, particularly obesity and insulin resistance [6,13].

Despite its recognized importance, our understanding of the precise mechanisms through which gut microbiota influences patients with PCOS remains incomplete [6,12,14,15,16]. A key function of the gut microbiome is the fermentation of dietary fibers into short-chain fatty acids (SCFAs), primarily including acetic acid (AA), propionic acid (PA), butyric acid (BA), and valeric acid (VA) [12,17,18]. Among these, BA is particularly noteworthy due to its dual function: it not only serves as a vital energy source for intestinal epithelial cells but also exhibits potent anti-inflammatory properties [19]. SCFAs collectively play a crucial role in maintaining the integrity of the intestinal mucosal barrier and act as significant signaling molecules within the gut ecosystem [20,21]. Furthermore, these fatty acids are integral to metabolic processes that enhance glucose-stimulated insulin secretion, thus improving insulin sensitivity. This enhancement is especially beneficial for individuals with PCOS, as improved insulin sensitivity helps regulate peptide hormones that control appetite and modulate overall metabolic health [22]. These insights emphasize the need for further research into the gut microbiota’s role in PCOS, suggesting potential therapeutic interventions that target microbial imbalances to alleviate the metabolic and inflammatory symptoms associated with the condition.

Metformin is well known for its metabolic effects [23,24], and exploring its potential impact on the gut microbiota and SCFA production represents a novel aspect of this investigation. By analyzing changes in SCFA profiles and and microbial composition following metformin therapy, we aspire to uncover the therapeutic mechanisms and potential benefits of metformin in the context of PCOS.

The aim of this study is to investigate the role of intestinal flora and its metabolites, particularly SCFAs, in the development of PCOS, and to assess the effects of metformin therapy on these components. The findings from this research may have significant implications for developing targeted interventions and personalized treatment strategies for women affected by PCOS, potentially leading to more effective and individualized therapeutic options.

## 2. Results and Discussion

### 2.1. Clinical Profiles of Study Subjects

In a study of 69 patients with PCOS, the mean body mass index (BMI) was 24.8 ± 5.2 kg/$m^{2}$, with 60.9% classified as normal weight, 20.3% as overweight, and 18.9% as obese; the control group’s BMI was slightly lower at 24.4 ± 4.8 kg/$m^{2}$ (p>0.05). Clinical hyperandrogenism was found in 66.7% of patients, while 78% exhibited biochemical hyperandrogenism, with elevated total testosterone in 53.6% and free testosterone in 34.8%. Morphological ultrasound signs of PCOS were evident in 87.0% of patients, with an average ovarian volume of 12.7 ± 3.2 ${\mathrm{cm}}^{3}$ and follicle count of 26.8 ± 18.8. Glucose tolerance tests indicated 11.6% had impaired glucose tolerance, alongside insulin resistance in 36.2% and hyperinsulinemia in 39.1%. Dyslipidemia was present in 29.0%, and dual-energy X-ray absorptiometry revealed excess total fat in 79.7% of patients, averaging 37.0 ± 7.5% total fat, including excessive visceral fat in 67.3% of those with excess fat, averaging 540.7 ± 482.9 g.

In patients with PCOS, the average serum levels of interleukin-6 (IL-6), tumor necrosis factor-alpha (TNF-α), and C-reactive protein (CRP) were significantly elevated compared to the control group, with respective levels of 1.3 pg/mL, 0.8 pg/mL, and 4.8 mg/L, while the control group recorded 0.7 pg/mL, 0.4 pg/mL, and 3.1 mg/L. Elevated levels of CRP and IL-6 were found in 37.7% of PCOS patients, but notably, no participants in either group exhibited increased TNF-α levels.

The mass of body fat, relative proportion of total fat, and mass of visceral fat have very strong direct associations with each other, as well as with the mass of android fat and BMI (r>0.80). Additionally, all these parameters have inverse associations with the level of high-density lipoprotein (HDL), indicated by negative correlation coefficients (in case of mass of body fat r=−0.55, p<0.001). Excluding the mass of total fat, these parameters also exhibit negative correlations with the levels of Sex Hormone-Binding Globulin (SHBG) and androstenedione. Furthermore, body fat parameters positively correlate with low-density lipoprotein (LDL), triglycerides, C-reactive protein, and fasting insulin, suggesting a strong link between body fat composition and various metabolic and cardiovascular risk factors (Figure 1, see Appendix A).

### 2.2. PCOS-Induced Disturbance in the Gut Microbiota

Recent scientific investigations have increasingly focused on the relationship between chronic low grade inflammation and metabolic dysfunction, particularly as they relate to disruptions in the gut microbiota composition. The complex interplay between microorganisms and the human immune system, along with the synthesis of biologically active substances by these microorganisms, plays a crucial role in the regulation of various metabolic processes. The gut microbiota is not only vital for digestion, energy balance, and maintaining the intestinal barrier function, but also influences fat storage, vasculature formation, the regulation of nervous and immune systems, and drug metabolism, among other functions [25,26]. A stable equilibrium between the gut microbiota and the host is essential for maintaining homeostasis and resistance against a range of pathologies.

The human gastrointestinal tract is home to a diverse community of bacterial species, often numbering in the hundreds. These bacteria share many functional roles, allowing for a degree of compensation through functional redundancy. This redundancy, however, poses challenges in identifying specific microorganisms that may play roles in the onset or progression of diseases like PCOS, which is associated with metabolic dysfunction. Reduced microbial diversity and functional redundancy have been strongly linked to conditions such as obesity, metabolic syndrome, and type 2 diabetes [27,28,29]. Numerous studies have demonstrated a correlation between decreased diversity of gut microbiota, insulin resistance, and obesity [30,31].

Alterations in gut microbial composition appear to significantly contribute to the pathogenesis of PCOS [10,13]. Individuals with PCOS exhibit reduced gut microbiota diversity, and current research indicates that the most prevalent genera in these individuals include *Clostridium* (phylum *Bacillota*), *Bacteroides* (phylum *Bacteriodota*), and *Escherichia* (phylum *Pseudomonadota*). While many members of the genera *Lactobacillus* and *Bifidobacterium* are beneficial symbionts that confer health advantages to the host, it is important to note that the genus *Clostridium* contains both pathogenic species-such as *C. tetani*, *C. botulinum*, *C. difficile*, and *C. perfringens*-as well as beneficial species, including the SCFA-producing group *C. leptum*, which encompasses *F. prausnitzii*, a dominant bacterial species in the large intestine [19].

A comprehensive quantitative analysis of gut microbiota composition has revealed significant differences in gastrointestinal tract colonization levels for key microorganisms, especially enzyme and SCFA producers (see Figure 2). In the PCOS group, a significant reduction in symbiotic bacteria was observed, particularly within the *C. leptum* group and *Prevotella* spp. This decline was especially pronounced in individuals experiencing increased body weight. Furthermore, levels of other beneficial microorganisms, including *A. muciniphila*, *F. prausnitzii*, *Bifidobacterium* spp., *Lactobacillus* spp., *Desulfovibrio* spp., and *Bacteroides* spp., were generally lower in the PCOS group compared to healthy controls.

The decrease in symbiotic bacteria was accompanied by an overgrowth of opportunistic microorganisms, which can impair tight junction protein expression and increase intestinal permeability [13,32]. This increased permeability facilitates the translocation of toxins—particularly lipopolysaccharides—into the systemic circulation, which activates the immune system and provokes chronic low grade inflammation [33]. The heightened presence of opportunistic pathogens, specifically those related to the *C. perfringens* group and *Staphylococcus* spp., observed in PCOS patients, may further contribute to chronic inflammation. Although weak positive correlations were noted between these opportunistic microorganisms and markers indicative of chronic inflammation, it is important to consider that the inflammation associated with PCOS is multifactorial and influenced by numerous intermediates that stimulate pro-inflammatory cytokine synthesis. Moreover, elevated titers of *C. difficile* and *Streptococcus* spp. were detected in PCOS patients, although these findings did not achieve statistical significance (p>0.05). In conclusion, the gut microbiota in individuals with PCOS is characterized by reduced species richness and a skewed microbial balance, with an increase in opportunistic pathogens and a decline in beneficial symbionts [13,32,34]. This microbial imbalance, compounded by genetic and epigenetic factors, may serve as an additional determinant in the development of metabolic disorders and obesity associated with PCOS.

### 2.3. SCFAs in Feces and Blood Serum (GC/MS)

Recent advances in the study of SCFAs, particularly AA, PA, BA, and VA, have elucidated their significant effects on various physiological systems at both cellular and molecular levels. Emerging research has demonstrated that the presence or deficiency of SCFAs can influence the pathogenesis of a wide array of diseases, including autoimmune disorders, metabolic syndromes, and neurological conditions [35].

In patients with PCOS, gut microbiota dysbiosis has been observed, which is characterized by an abnormal composition of microbial metabolites such as SCFAs [11,13,14]. These bacterial-derived molecules play a critical role in regulating food intake, energy expenditure, and maintaining intestinal immune homeostasis. Specifically, BA is predominantly utilized by colonocytes in the gut, whereas AA and PA are transported to the liver via the portal vein. In the liver, propionate is further metabolized by hepatocytes, while acetate often enters the peripheral circulation [35]. Research has shown that BA can stimulate the secretion of intestinal hormones such as glucagon-like peptide-1 (GLP-1) and peptide YY (PYY), which play crucial roles in reducing blood glucose levels by promoting insulin secretion and inducing a feeling of satiety [36,37]. Additionally, BA supports intestinal barrier function [38], enhances mucin production [39], and exhibits significant anti-inflammatory effects [40]. Therefore, a reduction in the population of bacteria that produce BA may contribute to the development of chronic low grade inflammation and insulin resistance. Acetate and PA also significantly impact metabolic processes. It is believed that acetate and propionate influence the function of pancreatic β-cells, enhancing insulin secretion and thus lowering blood glucose levels [41,42]. Given their profound roles in metabolic processes, SCFAs have been identified as crucial metabolic biomarkers in the context of PCOSs [17,19,22]. Alterations in the levels and utilization of these SCFAs could potentially shed light on the underlying mechanisms of PCOS, paving the way for novel diagnostic and therapeutic strategies. The identification of specific SCFA profiles associated with PCOS may lead to targeted interventions aimed at restoring healthy gut microbiota and SCFA production, thereby mitigating the metabolic and inflammatory disturbances characteristic of the condition. This could ultimately improve clinical outcomes for patients suffering from PCOS and related metabolic dysfunctions.

The investigation of gut microbiota metabolites in patients with PCOS has underscored its critical role in the development and progression of this condition (as illustrated in Figure 3 and detailed in Appendix A). Specifically, in the PCOS group, the levels of AA and VA were significantly elevated (p<0.05). PA and BA also showed an increasing trend, although these changes did not reach statistical significance (p>0.05). These findings align with the results reported by Li et al. (2022) [43]. Zhang et al. (2019) reported significantly greater fecal SCFAs in controls compared to women with PCOS [44]. Recent studies have confirmed that an energy-restricted diet significantly reduces the levels of fecal AA and BA in obese and overweight patients with PCOS, thereby improving their serum lipid profile [45,46,47]. Thus, the overproduction or accumulation of AA in the gut may contribute to the development of obesity and PCOS. In serum, we observed a non-significant (p>0.05) trend toward decreased levels of AA, PA, BA, and VA Appendix A). The SCFA levels in serum did not correlate with those in feces (Appendix A), indicating that the concentration of SCFAs in the bloodstream is influenced by a variety of interrelated factors. These factors include the permeability of the large intestine’s epithelial layer, the acidity levels in the colon, and variations in the metabolic activity of immune cells and colonocytes, which are the primary consumers of SCFAs. Additionally, genetic predisposition plays a role, as individual variations in the expression of SCFA transporters, MCT-1 and SMCT-1, by colonocytes can significantly impact SCFA concentrations in the serum [16,19]. These elements interact in complex, nonlinear ways to affect SCFA levels. In addition to this, a portion of the acetate present in the blood is of endogenous origin, being released by various tissues and organs [48].

PCOS is associated not only with reproductive disorders but also with metabolic abnormalities, and constitutes a known risk factor for type 2 diabetes and metabolic syndrome [1,2,3]. Roughly one-third of patients with PCOS exhibit laboratory evidence of low grade inflammation, such as elevated levels of CRP and/or IL-6. Given that most PCOS patients have excess adipose tissue, an evaluation of the levels of pro-inflammatory markers and adipocytokines in their serum was conducted, examining their relationship to body composition and serum SCFAs (Figure 4). The absolute concentration of serum AA was inversely associated with BMI, the mass of visceral fat, the relative proportion of total fat, TNF-α, triglycerides, and insulin levels one hour after glucose loading, as well as IL-6. The strongest correlations for AA were observed with pro-inflammatory TNF-α (r=−0.52, p=0.001) and IL-6 (r=−0.51, p=0.002). Additionally, AA concentration showed a positive association with SHBG (r=0.37, *p* = 0.03) and HDL (r=0.36, p=0.047) (Figure 4, Appendix A).

The presence of AA in the bloodstream potentially has a protective effect against PCOS. AA inversely correlated with key clinical manifestations of PCOS, including high BMI, insulin resistance, and systemic inflammation, indicating that elevated AA concentrations are associated with a lower severity of these pathological features. This underscores the potential therapeutic value of AA in managing PCOS. Moreover, the strong negative correlation between the primary serum SCFAs—AA (r=−0.52, p=0.001), PA (r=−0.46, p=0.004), BA (r=−0.49, p=0.002), and VA (r=−0.39, p=0.02)—and TNF-α—a critical player in the inflammatory and metabolic derangements observed in PCOS—confirms the beneficial effect of blood SCFAs in the pathogenesis of PCOS. This relationship highlights how SCFAs can mitigate the inflammatory processes associated with the condition. The primary mechanism through which SCFAs exert their anti-inflammatory effects is by inhibiting the synthesis of pro-inflammatory mediators such as IL-6 and TNF-α [49].

To elucidate the precise mechanisms by which SCFAs impact PCOS development and progression, extensive and detailed research is required. Future studies should focus on exploring the specific biochemical interactions, cellular responses, and broader physiological effects of SCFAs within the context of PCOS. This will provide clearer insights and potentially reveal novel therapeutic targets for managing this complex condition.

### 2.4. Prediction of Metformin Therapy’s Success

PCOS is a heterogeneous disease, which complicates the development of optimal treatment regimens. Treatment is symptomatic, and almost all prescribed drugs are used “off-label”, which limits the maximum efficacy and often reduces therapy compliance due to adverse events. Although combined oral contraceptives are commonly prescribed, insulin sensitizers, particularly metformin, are also widely used. Metformin’s beneficial effects on glucose metabolism are well-established: it reduces gluconeogenesis in the liver, decreases glucose absorption in the intestine, enhances peripheral glucose utilization, and increases the secretion of GLP-1 [23]. Additionally, it has been demonstrated that metformin can reduce the severity of chronic inflammation indirectly by improving metabolic parameters and directly due to its anti-inflammatory effects [24]. Considering identified disturbances in the intestinal microflora, the impact of metformin on the bacterial composition of the gut and the potential for enhancing therapy efficacy by correcting the microbiota with probiotics are of particular interest.

After six months of metformin therapy in the PCOS group (n=69), the intermenstrual interval decreased by 64.2 ± 46.1 days (p<0.001). Full restoration of the menstrual cycle was achieved in 16 out of 69 patients (23.2%). A partial improvement, characterized by an increase in the number of menstrual periods, was observed in 23 out of 69 patients (33.3%). However, oligo- or amenorrhea persisted in 30 out of 69 patients (43.5%), indicating no effect from the therapy.

Significant changes in body composition were observed during metformin therapy: the percentage of total adipose tissue decreased from 37.0 ± 7.3% to 35.4 ± 6.6% post-therapy (p<0.001), and the mass of visceral adipose tissue decreased from 654.4 ± 651.7 g to 382.5 ± 416.2 g (p=0.001). Among metabolic parameters, fasting insulin levels showed a statistically significant decrease of 27.1%, which was also reflected in a decrease in the HOMA index (p=0.006). However, lipid profile parameters did not change significantly (p>0.05). The number of patients with elevated CRP levels decreased by 1.5 times, while the incidence of elevated IL-6 levels decreased by 3.5 times. These results confirm the beneficial effect of metformin in PCOS.

In the study participants, metformin therapy not only reduced the frequency of insulin resistance and markers of chronic low grade inflammation, but also led to an increase in the abundance of symbiotic bacteria in gut microbiota, such as *A. muciniphila*, and a decrease in opportunistic pathogens such as *C. perfringens* and *C. difficile*. Metformin treatment demonstrates a selective effect on SCFA levels in feces, significantly reducing AA to normal levels, while not substantially affecting other SCFAs (Figure 3). Although the levels of SCFAs in the blood showed a persistent decrease following metformin treatment (Appendix A), this reduction did not reach statistical significance when compared to the control group (p>0.05). Thus, metformin’s impact on SCFAs may be more pronounced in the gastrointestinal tract than in the bloodstream. Further research is warranted to better understand the mechanisms behind these differential effects and their potential clinical implications in the management of conditions like PCOS. Given the lack of cycle regulation during metformin therapy in approximately half of the patients (43.5%), the cohort was further divided into groups with a full effect (n=16) and those with no effect (n=30). Significant differences in gut microbiota composition were observed before treatment between groups that exhibited a full effect from the therapy and those that did not (Figure 5). In patients who experienced the full effect of the therapy, the abundances of *C. leptum* gr., *Butyricimonas* spp., *Parabacteroides* spp., *Bacteroides* spp., and the combined group of *Dialister*, *Alisonella*, *Megaspherae*, and *Vellonella* were significantly higher than in patients who did not respond to the therapy (p<0.05). Most of these bacteria are considered beneficial for gut and metabolic health. Notably, the *C. leptum* gr. includes *F. prausnitzii*, one of the most abundant and important species in the human gut microbiota. A decrease in the number of these symbiotic bacteria may lead to reduced synthesis of SCFAs, which play critical roles in regulating glucose metabolism, fatty acid oxidation, and even appetite. Additionally, these bacteria are involved in the production of bile acids, which are essential for the digestion and emulsification of fats. A reduction in these symbionts could therefore contribute to the development of dyslipidemia and insulin resistance, conditions often associated with PCOS [50].

To predict the effectiveness of metformin therapy, specifically the complete restoration of the menstrual cycle as an indicator of treatment success, several eXtreme Gradient Boosting (xGBoost) [51] models were developed. A model, based on the levels of fecal SCFAs, the relative concentration of VA, and the absolute concentrations of AA and PA, demonstrated high accuracy (83%) and perfect sensitivity (100%) but a comparatively low specificity (71%) (Appendix A) (Table 1).

In another xGBoost model that considered various bacterial genera, species, families, and the fungal genus *Candida*, performance metrics improved significantly for predicting the full therapeutic effect (Appendix A). This model achieved an excellent accuracy of 93%, a sensitivity of 86%, and a specificity of 100% (Table 1).

When an xGBoost model combined levels of fecal SCFAs and microbiota features (Appendix A), it maintained a perfect sensitivity of 100%, but once again, specificity was lower at 71%.

Finally, utilizing the absolute serum concentrations of BA and VA, along with their relative concentrations, we developed another xGBoost model (Appendix A). This model achieved superior performance metrics, boasting a sensitivity of 100%, an accuracy of 91%, and a specificity of 80% (Table 1, Figure 6). Predicting the response to metformin therapy enables the optimization of treatment strategies for patients with PCOS. For individuals with an a priori unfavorable prognosis, it is advisable to consider alternative methods or combination therapies.

This study has several limitations that warrant consideration. Firstly, there is a significant imbalance in the sample sizes, with only 69 cases compared to 18 controls. Despite this disparity, the analysis yielded a high fold change and a medium Cohen’s d effect size for the potential PCOS markers, suggesting promising results. However, as this is a single-center study, it is crucial to expand the sample size and incorporate diverse ethnic groups in future research to enhance the generalizability of the developed models for predicting the effectiveness of metformin therapy.

Secondly, the analysis of SCFAs was limited to only four primary bacterial metabolites: AA, PA, BA, and VA. Future studies could benefit from an expanded panel that includes additional longer and branched-chain fatty acids to provide a more comprehensive understanding of SCFA profiles and their implications.

Lastly, the absence of a placebo group in this study presents another limitation. Including a placebo cohort in future trials would allow for more robust comparisons and conclusions regarding the efficacy and safety of the interventions being studied. Addressing these limitations in subsequent research will strengthen the validity of findings and contribute to a more nuanced understanding of PCOS and its associated treatments.

## 3. Materials and Methods

### 3.1. Study Design

The prospective study included patients who initially visited the scientific and outpatient department of the National Medical Research Center for Obstetrics, Gynecology and Perinatology named after Academician V.I. Kulakov in the period from July 2020 to February 2023. All participants provided signed informed consent.

The inclusion criteria for the PCOS group required individuals to demonstrate both clinical and laboratory signs of PCOS as determined upon their initial consultation at the Center. Participants needed to be within the age range of 18 to 35 years (25.3 ± 5.9). Conversely, individuals were excluded from the PCOS group if they had any androgen-secreting ovarian tumors or uncompensated endocrinological conditions, as well as congenital adrenal hyperplasia identified through genetic screening (CYP-21) or Cushing syndrome, and uncontrolled extragenital health issues. Furthermore, those who had utilized lipid-lowering medications, hormonal drugs, antibiotics, probiotics, or insulin sensitizers within the three months preceding their inclusion in the study were also disqualified. Other exclusion criteria encompassed pregnancy, gastroenterological diseases that could impact intestinal function, ongoing exacerbations of chronic inflammatory conditions from various origins, and any acute inflammatory diseases associated with the gastrointestinal tract or genitourinary system occurring less than three months prior to enrollment.

For the control group, participants had to be aged between 18 and 35 years (26.6 ± 5.0), maintaining a regular menstrual cycle without evidence of clinical hyperandrogenism. Exclusion criteria for the control group mirrored those of the PCOS group.

PCOS was diagnosed according to the clinical guidelines established by the Rotterdam consensus, which stipulate that at least two of three diagnostic criteria must be met. These criteria include oligo-anovulation, hyperandrogenism, and polycystic ovarian morphology. Oligo-anovulation is characterized by menstrual cycles exceeding 35 days or fewer than eight menstrual periods per year. Hyperandrogenism can be identified through various clinical signs, such as elevated testosterone levels or abnormal scores on the Ferriman–Gallwey hirsutism scale. Additionally, polycystic ovarian morphology is evaluated via transvaginal ultrasound, where the presence of 20 or more follicles in either ovary or an ovarian volume of 10 cm³ or greater indicates this criterion.

In alignment with clinical recommendations for the diagnosis of PCOS, a hormonal profile study was conducted on the 2nd or 3rd day of either a spontaneous or progesterone-induced menstrual cycle. Additionally, an ultrasound of the pelvic organs was performed between the 5th and 7th day of the menstrual cycle [52].

The levels of IL-6 and TNF-α in peripheral blood serum samples were quantified using a solid-phase enzyme immunoassay, employing test systems from “Vector-Best’’ (Koltsovo, Russia). The readings were obtained using the Infiniti F50 plate spectrophotometer (TECAN, Männedorf, Switzerland). Additionally, leptin and adiponectin levels were determined through solid-phase enzyme immunoassays utilizing commercial kits: “Leptin ELISA’’ (DBC, London, ON, Canada) for leptin, and “Human Adiponectin ELISA’’ (BioVendor, Brno, Czech Republic) for adiponectin. CRP levels were assessed in serum using the turbidimetric method on an automated analyzer (BA-400, Biosystems, Barcelona, Spain), with reagents specifically labeled “C-reactive protein (CRP)’’ (Biosystems, Spain).

Blood glucose and lipid profile analyses were performed using both photometric and turbidimetric methods on automated analyzer BA-400 (Biosystems, Spain). To diagnose carbohydrate metabolism disorders and assess insulin resistance, a 2 h oral glucose tolerance test was conducted with a 75 g glucose load. During this test, fasting glucose and immunoreactive insulin levels were measured, followed by subsequent measurements taken at 60 min intervals throughout the 2 h duration. The Homeostasis Model Assessment of Insulin Resistance (HOMA-IR) was calculated using the following formula: glucose (mmol/L) × insulin (µIU/mL)/22.5. An index value of HOMA-IR greater than 2.7 was used as a criterion for diagnosing insulin resistance, as established in previous studies [53]. Impairments in glucose tolerance were identified based on specific thresholds: a post-load glucose level exceeding 7.8 mmol/L indicated impaired glucose tolerance, while fasting glucose levels ranging from 6.1 to 7.0 mmol/L were also indicative of potential glucose metabolism issues.

Body composition was evaluated utilizing DXA with a Lunar 8743 device (GE Medical Systems, Madison, WI, USA). The assessment included the analysis of several key parameters: the percentage of total body fat, total body fat mass, trunk fat mass, and the android-to-gynoid fat ratio. A total body fat percentage of 30% or higher was considered indicative of excess body fat. Additionally, the volume and mass of visceral adipose tissue were determined using the “CoreScan’’ program by Corescan Pty Ltd. (Perth, Australia). An excess of visceral adipose tissue was diagnosed when the measured visceral fat mass exceeded 235 g [54]. Reference ranges for laboratory parameters are provided in Appendix A.

As a result, fecal and serum samples were collected from 87 patients, comprising 69 individuals diagnosed with PCOS and 18 individuals in the control group (Table 2).

All patients with PCOS (n=69) underwent a 6-month metformin therapy regimen, taking 1500 mg of “Glucophage-long’’ (Merck Sante, Lyon, France) daily. The therapeutic goal was identical for all included patients, in particular, the complete restoration of the menstrual cycle. The efficacy of the metformin treatment was categorized as follows: a full effect was indicated by the complete restoration of the menstrual cycle, a partial effect was marked by an increase in the frequency of menstrual periods, and no effect was denoted by the persistence of oligo- or amenorrhea (Table 3).

### 3.2. Collection and Storage of Samples

Fecal samples were collected in sterile containers with a volume of 8 to 10 cm^3^ and transported to the laboratory in a bag containing a gas-generating composition (AnaeroGen, ThermoScientific, Waltham, MA, USA) to create anaerobic conditions, following the provided instructions. Upon arrival at the laboratory, each sample was divided into two parts. The first portion was transferred into plastic Eppendorf tubes with a capacity of 1.5 to 2 cm^3^ and immediately frozen at −80 °C to maintain the integrity of the SCFAs. The second portion was reserved for microbiological analysis. Serum samples were obtained through venipuncture into vacutainer tubes without anticoagulants. Following collection, the tubes were centrifuged at 4000 rpm for 15 min to separate the serum from the blood cells. The resulting supernatant serum was carefully transferred to a labeled screw-cap tube and stored at −80 °C until further analysis.

### 3.3. GC/MS Quantitative Analysis of SCFAs

In this study, we employ a precise and sensitive method using Gas Chromatography with Mass Spectrometry (GC/MS) to quantify SCFAs, specifically AA, PA, BA, and VA, in the fecal and blood samples of women diagnosed with PCOS. The GC/MS technique is renowned for its ability to accurately measure these SCFAs, which are crucial indicators in metabolic studies [55,56,57]. We utilize a highly polar stationary phase column within the GC/MS system, which is particularly effective for the analysis of SCFAs without derivatization. We also applied liquid–liquid extraction [58,59] and acidification to the samples to ensure that the SCFAs were predominantly in their undissociated form, thereby increasing their hydrophobicity and volatility [60,61,62].

#### 3.3.1. Chemicals and Reagents

For the comprehensive GC/MS analysis of SCFAs in serum and feces samples obtained from women with PCOS and from the control groups, a variety of high-quality chemicals and reagents were employed, sourced from reputable suppliers such as Sigma-Aldrich, Saint Louis, MO, USA. The analysis included reference standards such as AA (≥99%), PA (≥99.5%), BA (≥99.5%), and VA (≥99%). Internal standards (IS) were utilized to enhance quantification accuracy: AA–d4 (≥99.5%) (AA*) was added for the quantification of serum AA, while BA–1,2–^13^$C_{2}$ (≥98%) (BA*) was used for the quantification of serum PA, BA, and VA, as well as fecal AA, PA, BA, and VA.

Milli-Q water was used for the reconstitution and dilution of samples during fecal sample preparation, while 1.0 M hydrochloric acid (HCl) was applied for sample acidification. Methyl tert-butyl ether (MTBE) served as the solvent for the liquid–liquid extraction (LLE) of SCFAs, ensuring effective extraction of the fatty acids from the biological matrices.

Working solutions of IS at appropriate concentrations were prepared and spiked into the samples to facilitate accurate quantification of SCFAs. All glassware, plasticware, and consumables specific to GC/MS analysis were rigorously cleaned and calibrated to maintain the integrity of the analytical process. To guarantee the accuracy and reliability of the results obtained in this study, all chemicals and reagents were meticulously handled in accordance with standard laboratory protocols.

#### 3.3.2. Preparation of Standard Solutions

Stock solutions of each SCFA were prepared in water. For PA, BA, and VA, the concentrations were set at 100 mM, while for AA, the concentration was 1000 mM. Additionally, working and calibration solutions of SCFAs were also prepared in water. IS solutions were prepared with precise concentrations to facilitate accurate measurements. For BA* IS, the concentrations were as follows: 15,000 µM for AA measurement in fecal samples, 1500 µM for the assessment of PA, BA, and VA in fecal samples, and 50 µM for determining the levels of PA, BA, and VA in serum. Meanwhile, AA* IS was formulated at a concentration of 500 µM to measure AA in serum accurately.

All solutions were stored at 4 °C. The stability of these solutions was assessed on a weekly basis, with results showing a relative standard deviation of less than 5%.

#### 3.3.3. Sample Preparation

Serum samples were thawed at 4 °C prior to analysis. Fecal samples were reconstituted by adding Milli-Q water at a ratio of 1000 µL per 50 mg of feces immediately before analysis. The fecal samples were then vortexed for 5 min to ensure proper mixing, followed by ultrasonic homogenization for 10 min and an additional 5 min of vortexing. The homogenized feces were centrifuged at 15,000 rpm for 5 min to separate the supernatant. The concentration of SCFAs in feces was measured in micromoles (µM) per 50 g of unprocessed feces.

#### 3.3.4. Extraction Procedure

A volume of 100 µL from standard solutions, serum samples, or supernatants derived from the homogenized feces was transferred into 500 µL plastic tubes. Following this, 10 µL of 1.0 M HCl was added to each sample to acidify the solution. In order to facilitate the quantification of SCFAs, 10 µL of the IS working solution was then spiked into each sample. The resulting sample mixtures were vortexed for 1 min, centrifuged at 15,000 rpm for 5 min to effectively separate the phases.

A volume of 100 µL of the resulting supernatants was transferred into new 500 µL plastic tubes. To initiate LLE, 200 µL of MTBE was added to each tube. The LLE process was activated by vigorously vortexing the mixture for 20 min, ensuring thorough and efficient extraction of the target compounds. Then, the tubes were centrifuged at 15,000 rpm for 5 min to promote phase separation. A volume of 100 µL from the MTBE phase, containing the extracted SCFAs, was transferred into autosampler vials fitted with glass inserts. Finally, the samples were subjected to analysis using GC/MS for the quantification of SCFAs.

#### 3.3.5. GC/MS Parameters

The sample analysis was conducted using an Agilent 7890B gas chromatograph coupled with an Agilent 5977B GC/MSD single quadrupole mass spectrometer (Agilent, Santa Clara, CA, USA), employing an HP–FFAP column (25 m length, 0.32 mm diameter, 0.5 µm film thickness) for chromatographic separation. Inlet temperature was set to 250 °C, and the injection volume was 1 µL. Two separate GC/MS conditions were utilized for investigating fecal and blood samples. For feces, the system operated in split mode with a ratio of 10:1. In contrast, for serum, a splitless mode was employed, with a switch to septum purge split mode and a purge flow to split vent of 100 mL/min at 0.1 min. Helium gas with a purity of ≥99.9999% was used as the carrier gas at a constant flow rate of 1.5 mL/min, with a septum purge of 3 mL/min. The GC oven temperature program was optimized for efficient separation of SCFAs: an initial temperature of 60 °C held for 1.5 min, increased by 40 °C/min to 100 °C, held for 1 min, further increased by 10 °C/min to 158 °C, and finally raised by 100 °C/min to 240 °C. The post-run time was specified at 240 °C. The transfer line, ion source, and quadrupole temperatures were maintained at 250 °C, 230 °C, and 150 °C, respectively. Electron ionization with an energy of 70 eV was employed to ensure efficient ionization of the analytes.

During the development of the separation and identification method, an approach described in a previous study outlined in a recent study published by Kim et al. (2022) was utilized [56]. This approach was tailored to accommodate the specific characteristics of the analyzed samples and the available materials and analytical standards. During the optimization of the GC/MS method, MS data acquisition was performed in full scan mode over the *m*/*z* range of 39–87. The choice of conditions and compound identification were guided by the injection of chemical standards. Retention times and corresponding mass spectra were compared, ensuring accurate identification of the analytes. The conditions were fine-tuned to maximize the signal-to-noise ratio, effectively separating the peaks of SCFAs from other volatiles present in the samples.

Finally, quantification of analytes was carried out in selected ion monitoring (SIM) mode using target ions (*m*/*z* 60.0—AA, BA, VA; *m*/*z* 62.0—BA*; *m*/*z* 63.0—AA*; *m*/*z* 74.0—PA) and confirmed by confirmative ions (*m*/*z* 43.0—AA; *m*/*z* 45.0—PA; *m*/*z* 46.0—AA* *m*/*z* 73.0—BA, VA; *m*/*z* 75.0—BA*). The integration of compounds was based on specific *m*/*z* values. Data acquisition and analysis were conducted using the Masshunter quantitative program, facilitating the processing and interpretation of chromatographic and mass spectral data.

#### 3.3.6. Calibration

Two series of calibration solutions were prepared from stock solutions of SCFAs immediately before calibration to quantification of SCFAs in feces. The first series comprised solutions of AA ranging from 100 to 3500 µM, while the second series included solutions of mixtures of PA, BA, and VA, each ranging from 10 to 350 µM. Subsequently, 10 µL of BA* IS (at the 1500 µM and 50 µM, respectively) and 10 µL of 1M HCl were added. Seven concentration levels (calibration curves: Appendix A) of calibration standards and four levels of quality control (QC), diluted in water, were prepared and extracted as described in the extraction procedure. The calibration levels for AA quantification were 100, 500, 1000, 1500, 2000, 2750, and 3500 µM. The QC levels for AA quantification were 500, 1500, 2000, and 3500 µM. The calibration levels for the PA–BA–VA–mixture quantification were 10, 50, 100, 150, 200, 275, and 350 µM of each component consequently. The QC levels for the PA–BA–VA–mixture quantification were 50, 150, 200, and 350 µM of each component consequently. The example chromatograms for the blank, calibration levels and the samples are presented in Appendix A.

For the quantification of SCFAs in serum, solutions were prepared with concentrations ranging from 0.2 to 10 µM for PA, BA, and VA, and from 2 to 100 µM for AA. Then, 5 µL of AA* and 5 µL of BA* ISs, along with 1 M HCl, were added. Eight concentration levels (calibration curves: Appendix A) of calibration standards and four levels of QC, diluted in water, were prepared and extracted as described in the extraction procedure. The calibration levels were 0.2, 0.6, 1, 2, 4, 6, 8, and 10 µM. The QC levels were 0.2, 2, 6, and 10 µM. The chromatograms for the blank, calibration levels and the samples are presented in Appendix A.

The calibration curves were constructed by plotting the peak area ratio of each SCFA to the corresponding IS against the concentration of each SCFA, followed by linear regression. The linearity of the calibration curve for each SCFA was assessed by the coefficient of determination (R^2^) value exceeding 0.99. The limit of detection (LOD) was calculated as 3.3 × SD/b, where SD is the standard deviation of the Y-intercept, and b is the slope of the linear regression curve. The limit of quantification (LOQ) was calculated as 3 × LOD.

### 3.4. Fecal Microbiota Analysis

DNA of intestinal-associated microorganisms (see Appendix A) were detected in fecal samples by real-time PCR. For this purpose, DNA extraction was performed using the Proba-Cito reagent kit (DNA-Technology LLC, Moscow, Russia). Subsequent analysis was conducted employing the Enterflor kit (DNA-Technology LLC, Russia), which is specifically designed for the evaluation of the most important members of the gut microbiota.

### 3.5. Statistical Analysis

SCFAs derived from serum and fecal samples, along with microbiota features, were compared between the PCOS group and the control group using the Mann–Whitney test, with a significance threshold set at p<0.05. Signatures exhibiting statistically significant alterations were further assessed for correlation using the Spearman test, also with a significance threshold of p<0.05.

Clinical parameters were assessed for associations with SCFAs derived from feces and serum, as well as microbiota features, using the Spearman test with a significance level set at p<0.05. Additionally, body fat mass, total fat mass, the relative composition of total fat, and the mass of visceral fat were evaluated for associations with biochemical parameters. The associations between SCFAs from feces and fecal microbiota were also analyzed using the Spearman test, with a significance level of p<0.05.

Fecal and serum SCFAs and fecal microbiota data were used to create a diagnostic model for predicting the full success of treatment, specifically the restoration of the menstrual cycle. XGBoost models were developed through an iterative process where features were step-by-step excluded, provided that such exclusion improved model accuracy. Sensitivity and specificity were calculated using leave-one-out cross-validation, with the optimal threshold determined by maximizing the sum of sensitivity and specificity.

Data processing was performed using a laboratory-created script in R version 4.3.2, employing the XGBoost package version 1.7.6.1 [51] for model creation.

## 4. Conclusions

This study provides compelling evidence that the gut microbiota plays a pivotal role in the pathogenesis of PCOS. We have highlighted significant shifts in microbial composition among women with PCOS, marked by a reduction in beneficial bacteria associated with gut and metabolic health, alongside an overgrowth of opportunistic pathogens.

Notably, the imbalance in SCFA production, particularly the increased levels of AA and VA, aligns with previous studies and suggests a complex interplay between gut microbiota and obesity in the context of PCOS. While serum SCFA levels did not show a significant correlation with fecal concentrations, the presence of AA in serum may play a protective role against key symptoms such as high body mass index, insulin resistance, and chronic low-grade inflammation, underscoring its potential therapeutic relevance.

Metformin therapy proved to be beneficial, leading to significant improvements in endocrine and metabolic parameters, as well as notable changes in gut microbiota composition. The treatment not only normalized fecal AA levels but also restored gut microbiota. Our predictive model, based on serum concentrations of BA and VA, demonstrated considerable sensitivity and accuracy in forecasting the effectiveness of metformin therapy. Personalized treatment strategies for PCOS could incorporate dietary modifications and symbiotic supplementation in patients with less favorable microbiota profiles.

In summary, our findings illuminate the intricate relationship between gut microbiota, SCFAs, and the pathophysiology of PCOS. The identification of microbial imbalances and the implications of SCFA alterations offer critical insights that can inform therapeutic strategies. Future research is essential to elucidate the underlying mechanisms driving these relationships and to optimize treatment protocols, ultimately enhancing clinical outcomes for women affected by this complex condition.

## Figures and Tables

**Figure 1 ijms-25-10636-f001:**
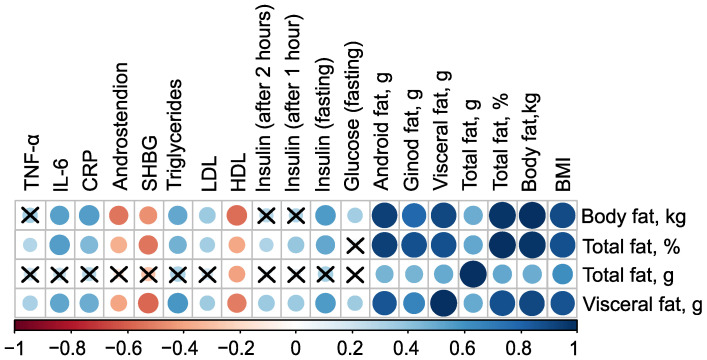
Correlation plot of body composition and clinical parameters in the control (n=18) and PCOS (n=69) groups before therapy. Cross sign marks statistically insignificant (p>0.05) relationship. The size of the dots associate with absolute value of correlation coefficient: stronger relationship—larger dots.

**Figure 2 ijms-25-10636-f002:**
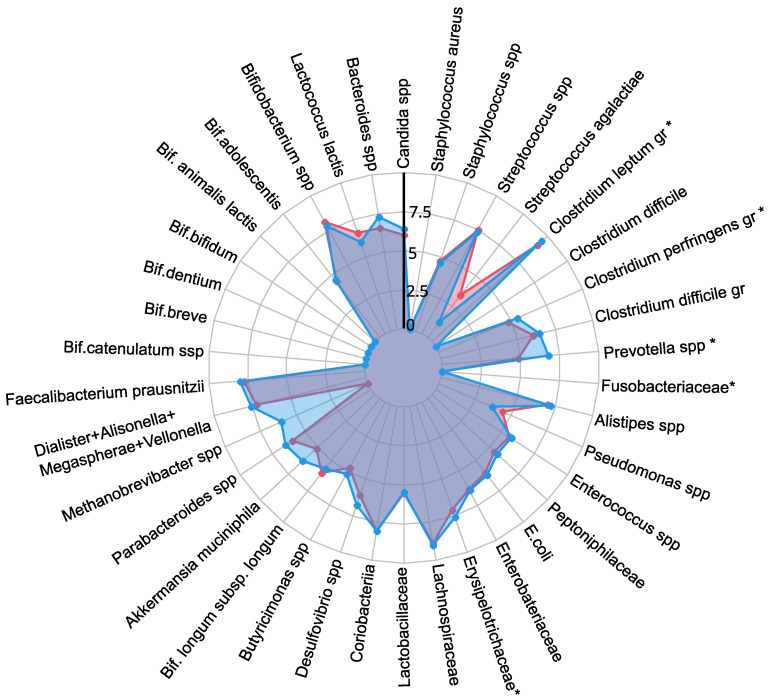
The influence of PCOS on gut microbiota before treatment, presented as a a radar chart (logarithmic scale) with median values of the number of colony-forming units: PCOS (red color, n=69) and control group (blue color, n=18). The data represent the average estimate of the $\log_{10}$ of fecal real-time polymerase chain reaction (PCR) target genetic amplicon copy numbers in 1 g of feces. Statistically significant alterations (p<0.05) are indicated by an asterisk (*).

**Figure 3 ijms-25-10636-f003:**
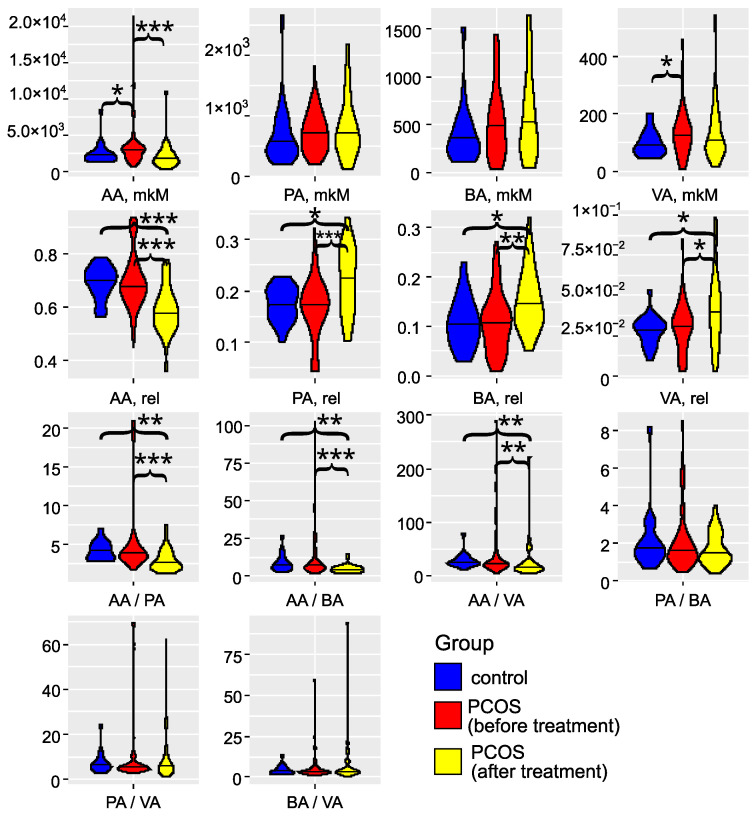
Violin plots for SCFA concentrations, relative abundances, and ratios (feces). Blue: the control group (n=18); red: the group with PCOS before treatment (n=69); yellow: the group with PCOS after treatment (n=32). AA: acetic acid; VA: valeric acid; BA: butyric acid; PA: propionic acid. * denotes p<0.05, ** denotes p<0.01, and *** denotes p<0.001, according to the Mann–Whitney test.

**Figure 4 ijms-25-10636-f004:**
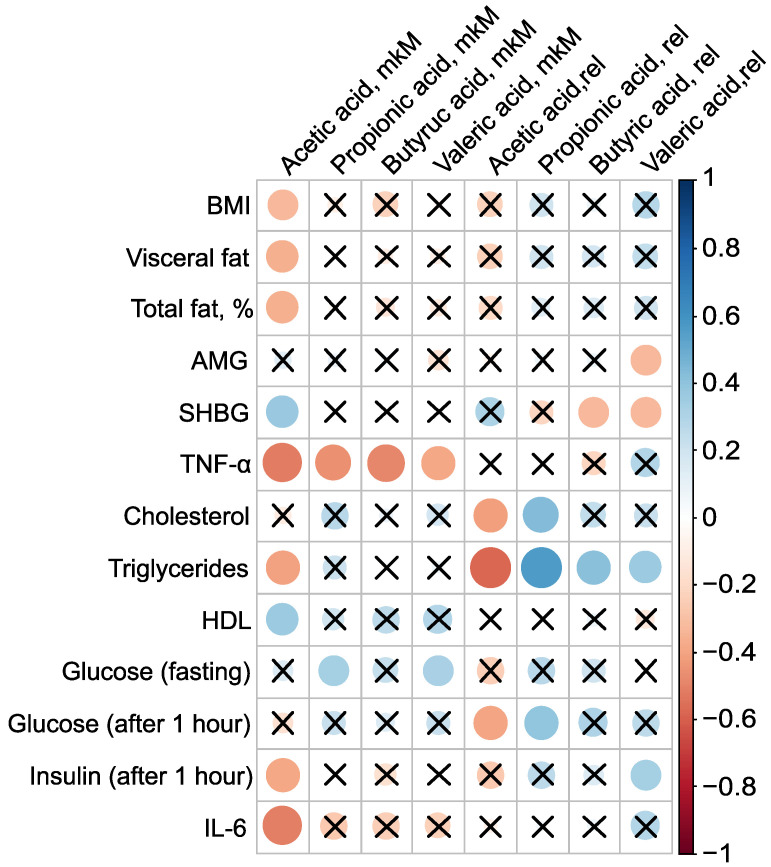
Clinically relevant parameters in the PCOS group (n=38) and their statistically significant associations with serum SCFAs before metformin therapy. Cross sign marks statistically insignificant (p>0.05) relationship. The size of the dots associate with absolute value of correlation coefficient: stronger relationship—larger dots.

**Figure 5 ijms-25-10636-f005:**
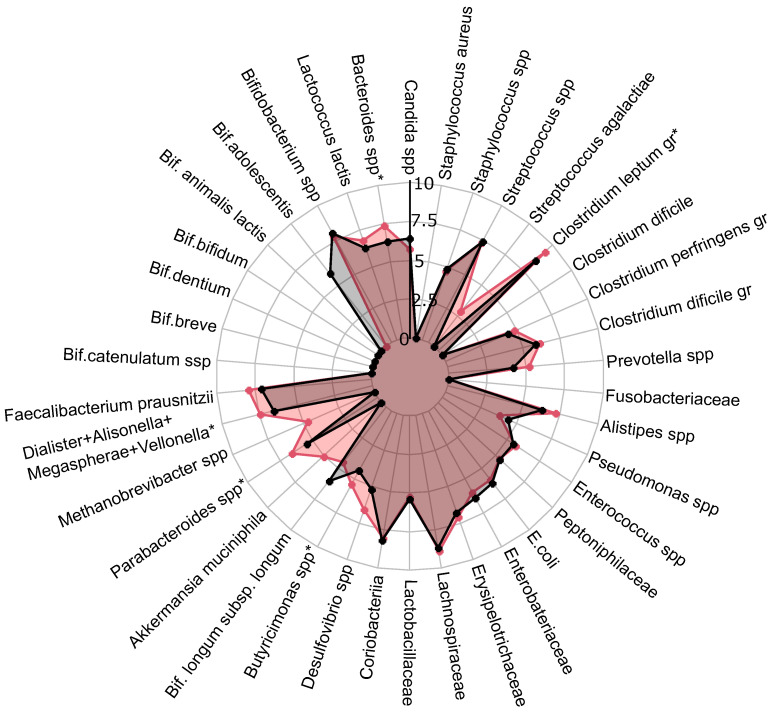
PCOS group gut microbiota composition before treatment with expected effect from therapy. Red: radar chart for full effect of therapy; black: no effect of therapy. Statistically significant alterations (p<0.05) are indicated by an asterisk (*).

**Figure 6 ijms-25-10636-f006:**
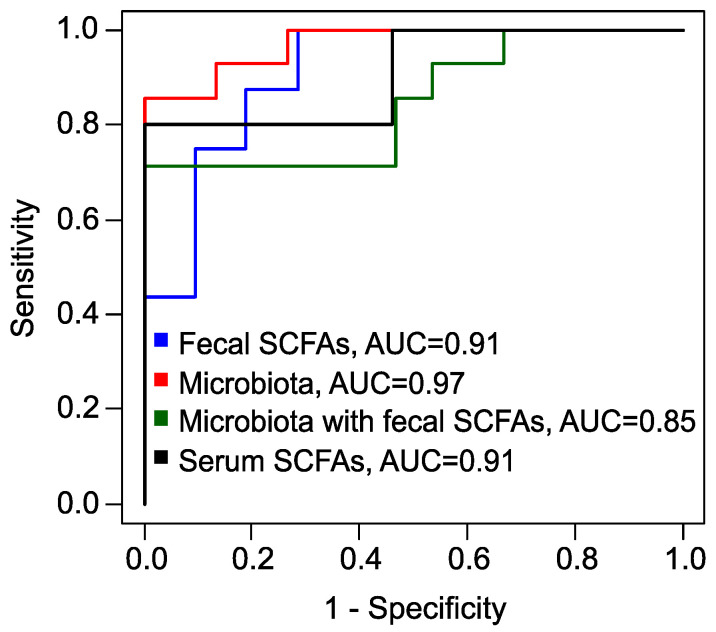
Receiver operating characteristic (ROC) curves, obtained by leave-one-out cross-validation of XGBoost models to predict the full effect of metformin therapy. Blue: model based on fecal SCFAs; red: based on fecal microbiota; green: combined data of fecal SCFAs and microbiota; black: based on serum SCFAs. AUC: area under the curve.

**Table 1 ijms-25-10636-t001:** Performance metrics of XGBoost models: Accuracy, sensitivity, specificity, and optimal threshold.

	Accuracy, %	Sensitivity, %	Specificity, %	Threshold
Fecal SCFAs	83	100	71	0.45
Gut microbiota	93	86	100	0.36
Gut microbiota & fecal SCFAs	86	100	71	0.30
Serum SCFAs	91	100	80	0.41

**Table 2 ijms-25-10636-t002:** Number of samples (feces and serum) obtained for the control group (n=18) and the PCOS group (n=69).

Type of Sample	Control (*n* = 18)	PCOS (*n* = 69)
Feces	18	69
Serum	10	38

**Table 3 ijms-25-10636-t003:** Number of samples (feces and serum) obtained for the PCOS group (*n* = 69) after metformin’s therapy according to the effect achieved.

Type of Sample	Full Effect (*n* = 16)	No Effect (*n* = 23)	Partial Effect (*n* = 30)
Feces	8	7	17
Serum	16	23	30

## Data Availability

The data supporting the reported results can be found at Appendix A.

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
