# Peer review of "Impact of Gut Microbiota and SCFAs in the Pathogenesis of PCOS and the Effect of Metformin Therapy"

_ijms, 2024, doi:10.3390/ijms251910636_

Round 1

Reviewer 1 Report

Comments and Suggestions for Authors

The authors of this study address the ever-present and important topic of the importance of the gut microbiome in women with PCOS in terms of the impact and contribution of metabolites to the pathogenesis of the syndrome. The study also covers the use of metformin therapy in patients with PCOS. There are several issues that need to be resolved before an article is deemed suitable for publication in a journal. The article needs a major overhaul as it should be more structured. Also, the topic of the article should be clearer. The abstract section should have a more structured format, in the sense that it should contain all the necessary sections instead of a narrative presentation, plus the methodology, parameters of the results and figures. The paper should have a separate introduction, clearly presenting the topic presented. The introduction should be separate from the objectives of the work, the methods used, etc., as this will be presented in subsequent sections of the paper. Hence, currently the Introduction section does not properly introduce the topic of the project. The materials and methods section should follow the Introduction. Materials and methods should describe the study and control group, with their inclusion and exclusion elements (which are currently missing), how patients were selected for the presented study and the research methods used by the authors of the paper, without presenting data from other studies. In the presented paper, the results are also not extracted, but already included in the discussion section. The discussion should include a more structured format, including direct comparisons of the exact results with relevant articles in the literature, together with the limitations of the study.

The final conclusions should reflect the exact results along with the quality of the study. In addition, clearer explanations of Figure 1 and Figure 4 should be provided.

Comments on the Quality of English Language

No comments

Author Response

Dear Editors and Reviewers,

Thank you for the constructive comments and questions concerning our manuscript. We have revised the manuscript accordingly and hope it meets your requirements. A list of point-by-point responses to the reviewers’ remarks is given below, along with a clear indication of the location of each revision.

------------------------------------------ Responses to Reviewer 1 ------------------------------------------

The authors of this study address the ever-present and important topic of the importance of the gut microbiome in women with PCOS in terms of the impact and contribution of metabolites to the pathogenesis of the syndrome. The study also covers the use of metformin therapy in patients with PCOS. There are several issues that need to be resolved before an article is deemed suitable for publication in a journal.

Comment: The article needs a major overhaul as it should be more structured. Also, the topic of the article should be clearer.

Response: Thank you for reviewing our article and providing valuable comments. We have addressed and corrected all the issues you identified.

Comment: The abstract section should have a more structured format, in the sense that it should contain all the necessary sections instead of a narrative presentation, plus the methodology, parameters of the results and figures.

Response: The abstract was reorganized to include all the necessary sections. Narrative descriptions in the aim and results were replaced. Parameters of the results fold change (FC), r2 and p-value were added.

Comment: The paper should have a separate introduction, clearly presenting the topic presented. The introduction should be separate from the objectives of the work, the methods used, etc., as this will be presented in subsequent sections of the paper. Hence, currently the Introduction section does not properly introduce the topic of the project.

Response: The the objectives of the work was clarified and replaced at the end of Introduction section. The description of methods used was replaced to the Material and Methods section.

Comment: The materials and methods section should follow the Introduction. Materials and methods should describe the study and control group, with their inclusion and exclusion elements (which are currently missing), how patients were selected for the presented study and the research methods used by the authors of the paper, without presenting data from other studies.

Response: This article was structures following the instructions of IJMS journal: “Materials and methods” section should follow the “Results and Discussion” section (https://www.mdpi.com/journal/ijms/instructions). “Materials and methods” section was corrected to include the inclusion and exclusion criteria of PCOS and control groups.

Comment: In the presented paper, the results are also not extracted, but already included in the discussion section. The discussion should include a more structured format, including direct comparisons of the exact results with relevant articles in the literature, together with the limitations of the study.

Response: We have combined sections 2 and 3 into one “Results and Discussion” (the rules of the MDPI: https://www.mdpi.com/journal/ijms/instructions), which is what we did for a structured narrative in the article. Here is an example of a recent article in this journal with the same sections and formatting rules: https://www.mdpi.com/1422-0067/25/13/7482. The limitations of the study were added at the end of the “Results and Discussion” section.

Comment: The final conclusions should reflect the exact results along with the quality of the study. In addition, clearer explanations of Figure 1 and Figure 4 should be provided.

Response: The conclusions were clarified. The explanations of Figure 1 and Figure 4 were corrected.

Reviewer 2 Report

Comments and Suggestions for Authors

manuscript very interesting. The growing problem of polycystic ovary syndrome is still being analyzed today, and we are still asking ourselves what leads to it and how to treat and accurately diagnose it. Of course, analysis of the gut microbiota , immune response, hormonal analysis, insulin and lipid paramtrs are some of the basic elements to start the diagnosis in addition to the routine ultrasound examination. Any treatment that improves parameters in PCOS are the hope of patients to improve their well-being but also most importantly to equalize parameters and exponents past in PCOS. For a long time there have been attempts to administer metformin, which gives great success of therapy and improvement of laboratory parameters. Paying attention to the intestinal microbiome ist reduction of unfavorable pathogens. Metformin alone, however, will not cause this. There should also be supplementation with symbiotics with properly selected Lactobacillus and a well-chosen diet.

Literature sufficient. Graphics presented interestingly and encouraging for its analysis.

Author Response

Dear Editors and Reviewers,

Thank you for the constructive comments and questions concerning our manuscript. We have revised the manuscript accordingly and hope it meets your requirements. A list of point-by-point responses to the reviewers’ remarks is given below, along with a clear indication of the location of each revision.

manuscript very interesting. The growing problem of polycystic ovary syndrome is still being analyzed today, and we are still asking ourselves what leads to it and how to treat and accurately diagnose it. Of course, analysis of the gut microbiota , immune response, hormonal analysis, insulin and lipid paramtrs are some of the basic elements to start the diagnosis in addition to the routine ultrasound examination. Any treatment that improves parameters in PCOS are the hope of patients to improve their well-being but also most importantly to equalize parameters and exponents past in PCOS. For a long time there have been attempts to administer metformin, which gives great success of therapy and improvement of laboratory parameters. Paying attention to the intestinal microbiome ist reduction of unfavorable pathogens. Metformin alone, however, will not cause this. There should also be supplementation with symbiotics with properly selected Lactobacillus and a well-chosen diet.

Literature sufficient. Graphics presented interestingly and encouraging for its analysis.

Response: Thank you for your kind opinion of our manuscript. We added alternative or adjunctive approaches in Conclusions.

Reviewer 3 Report

Comments and Suggestions for Authors

My comments on the submitted manuscript:

Abstract: Please avoid using overly subjective adjectives in the abstract (e.g., "intricate interplay" or "pivotal role").

Introduction: The introduction begins with the statement that "polycystic ovary syndrome (PCOS) is not fully understood," without providing definitions, diagnostic criteria, phenotypes, or an explanation of the short- and long-term consequences (e.g., metabolic). This weakens the justification for the submitted research. An example of a well-structured introduction for this type of study can be found in PMID: 35010948.

Additionally, the microbiome is not clearly defined and is limited to the gut microbiome. It should be mentioned that the microbiota relevant to PCOS are also located in the reproductive tract.

Methods/Results: These fundamental deficiencies make it difficult to follow the definition of the study cohort. What criteria were used for diagnosing PCOS? How was "clinical hyperandrogenism" defined and quantified? What cutoff values were used for laboratory investigations? What sonographic criteria were applied for PCO morphology? Were the therapeutic goals identical for all included patients (e.g., normalization of metabolic disturbances, ovulation, normoandrogenemia, pregnancy)?

Furthermore, how can the unbalanced ratio between cases (n=69) and controls (n=18) be explained? To obtain statistically meaningful results, the control group should be at least comparable in size or, preferably, larger. Was statistical power assessed?

Discussion: The discussion section (here labeled as "conclusions") is brief and confusing. It presents the study objectives ("Our objective is to elucidate the potential benefits of metformin in managing PCOS") but does not adequately discuss the results in relation to the findings. Additionally, several relevant studies are omitted from the discussion (see Chadchan et al. 2022, PMID: 35900833 for a review of recent studies).

Author Response

Dear Editors and Reviewers,

Thank you for the constructive comments and questions concerning our manuscript. We have revised the manuscript accordingly and hope it meets your requirements. A list of point-by-point responses to the reviewers’ remarks is given below, along with a clear indication of the location of each revision.

My comments on the submitted manuscript:

Comment: Abstract: Please avoid using overly subjective adjectives in the abstract (e.g., "intricate interplay" or "pivotal role").

Response: The abstract has been revised to eliminate subjective adjectives, ensuring a more objective and neutral tone.

Comment: Introduction: The introduction begins with the statement that "polycystic ovary syndrome (PCOS) is not fully understood," without providing definitions, diagnostic criteria, phenotypes, or an explanation of the short- and long-term consequences (e.g., metabolic). This weakens the justification for the submitted research. An example of a well-structured introduction for this type of study can be found in PMID: 35010948.

Response: We corrected the section “Introduction” to include the definitions, diagnostic criteria, phenotypes, or an explanation of the short- and long-term consequences definitions, diagnostic criteria, phenotypes, or an explanation of the short- and long-term consequences of PCOS.

Comment: Additionally, the microbiome is not clearly defined and is limited to the gut microbiome. It should be mentioned that the microbiota relevant to PCOS are also located in the reproductive tract.

Response: The section “Introduction” has been improved accordingly.

Comment: Methods/Results: These fundamental deficiencies make it difficult to follow the definition of the study cohort. What criteria were used for diagnosing PCOS? How was "clinical hyperandrogenism" defined and quantified? What cutoff values were used for laboratory investigations? What sonographic criteria were applied for PCO morphology? Were the therapeutic goals identical for all included patients (e.g., normalization of metabolic disturbances, ovulation, normoandrogenemia, pregnancy)?

Response: We have added the necessary information regarding the study design in the "Materials and Methods" section, in accordance with your recommendations. We added reference ranges for laboratory parameters in the Table S5 (Supplementary_S1).

Comment: Furthermore, how can the unbalanced ratio between cases (n=69) and controls (n=18) be explained? To obtain statistically meaningful results, the control group should be at least comparable in size or, preferably, larger. Was statistical power assessed?

Response: Thank you for your valuable comment. This limitation of the study were included in the “Results and discussion section”: “This study has several limitations that warrant consideration. Firstly, there is a significant imbalance in the sample sizes, with only 69 cases compared to 18 controls. Despite this disparity, the analysis yielded a high fold change and a medium Cohen's d effect size for the potential PCOS markers, suggesting promising results. However, as this is a single-center study, it is crucial to expand the sample size and incorporate diverse ethnic groups in future research to enhance the generalizability of the developed models for predicting the effectiveness of metformin therapy.”

Comment: Discussion: The discussion section (here labeled as "conclusions") is brief and confusing. It presents the study objectives ("Our objective is to elucidate the potential benefits of metformin in managing PCOS") but does not adequately discuss the results in relation to the findings. Additionally, several relevant studies are omitted from the discussion (see Chadchan et al. 2022, PMID: 35900833 for a review of recent studies).

Response: According to the MDPI rules, it is possible to combine two sections into one “Results and Discussion”, which is what we did for a structured narrative in the article. The article Chadchan et al. 2022 was added to this section and a comparative analysis of the data we obtained was conducted with the data in this article. We have streamlined the “Conclusions” section to ensure it is concise and clear. The revised version succinctly summarizes the key findings and implications of the study, avoiding unnecessary complexity. Each point is presented in a straightforward manner, facilitating a better understanding of the overall conclusions without ambiguity.

Reviewer 4 Report

Comments and Suggestions for Authors

The study titled Impact of Gut Microbiota and SCFAs in PCOS Pathogenesis and Metformin on Therapy aims to establish a relationship between GUT microbiota and PCOS.

This clinical study is very interesting and has merit. Some points should be further clarified.

1. The study hypothesis must be clearly presented in the introduction. The last paragraphs are confusing, presenting objectives and descriptions of the treatment. They must be straightforward in the hypothesis.

2. The methods must be better presented. It is not clear when the assessments were carried out.

Did the authors evaluate participants before and after treatment?

Were they only evaluated afterwards?

How was the sample chosen?

How were the controls chosen?

Patients treated with metiformin used only this drug?

Was there a placebo group?

Do the microbiota graphs show before and after treatment? Was there this comparison or was the comparison control and PCOS?

The results need to be better presented as they are confusing in relation to the groups and the time studied.

The discussion is good, but there should be more depth on future perspectives highlighted by the study.

Author Response

Dear Editors and Reviewers,

Thank you for the constructive comments and questions concerning our manuscript. We have revised the manuscript accordingly and hope it meets your requirements. A list of point-by-point responses to the reviewers’ remarks is given below, along with a clear indication of the location of each revision.

The study titled Impact of Gut Microbiota and SCFAs in PCOS Pathogenesis and Metformin on Therapy aims to establish a relationship between GUT microbiota and PCOS.

This clinical study is very interesting and has merit. Some points should be further clarified.

Comment: The study hypothesis must be clearly presented in the introduction. The last paragraphs are confusing, presenting objectives and descriptions of the treatment. They must be straightforward in the hypothesis.

Response: We corrected the section “Introduction”.

Comment: The methods must be better presented. It is not clear when the assessments were carried out.

Did the authors evaluate participants before and after treatment? Were they only evaluated afterwards? How was the sample chosen?

Response: Thank you for valuable remarks. The participants were evaluated before and after treatment.

We improved the description of Study design in Materials and methods section to clarify this point.

Comment: How were the controls chosen?

Response: The inclusion and exclusion criteria were added in Materials and methods section.

Comment: Patients treated with metiformin used only this drug?

Response: Metformin was the only drug used for all PCOS patients. This information was added in Materials and methods section.

Comment: Was there a placebo group?

Response: There was no placebo group. This limitation of the study were included in the “Results and discussion section”: “Lastly, the absence of a placebo group in this study presents another limitation. Including a placebo cohort in future trials would allow for more robust comparisons and conclusions regarding the efficacy and safety of the interventions being studied. Addressing these limitations in subsequent research will strengthen the validity of findings and contribute to a more nuanced understanding of PCOS and its associated treatments.”

Comment: Do the microbiota graphs show before and after treatment? Was there this comparison or was the comparison control and PCOS?

Response: We corrected the annotation of the Figures.

Comment: The results need to be better presented as they are confusing in relation to the groups and the time studied.

Response: The results in the "Results and Discussion" section have been presented more clearly, with a stronger emphasis on the specific groups and time frames studied.

Comment: The discussion is good, but there should be more depth on future perspectives highlighted by the study.

Response: Thank you for your positive feedback on our study. We have incorporated future perspectives into both the "Results and Discussion" and "Conclusions" sections. This addition aims to provide greater context and direction for subsequent research, highlighting the implications of our findings and potential avenues for further investigation.

Round 2

Reviewer 1 Report

Comments and Suggestions for Authors

The authors have revised their research paper. The research addresses the important topic of the importance of the gut microbiome in women with PCOS in terms of the impact and contribution of metabolites to the pathogenesis of the syndrome. Furthermore, the study also includes an analysis of the use of metformin therapy in patients with PCOS. However, a few issues remain to be systematised. The Materials and Methods section should follow the Introduction. This is where the study group, which is currently in the Results section (Section – Study Participants), and the control group should be described. Materials and methods only appear after the results and discussion, without the study group being described separately. Although the authors have completed several references, they could still be supplemented in the Discussion section, which, combined with the results, has a small stock of relevant articles in the literature.

Comments on the Quality of English Language

No comments.

Reviewer 3 Report

Comments and Suggestions for Authors I appreciate the efforts of the authors to improve the submitted manuscript. Most of the suggestions (except for the still too simple "Conclusions") have been addressed in a satisfactory manner, resulting in a significantly more informative manuscript.

Reviewer 4 Report

Comments and Suggestions for Authors

Dear Authors,

The new version with possible changes must be identified and marked in red or another to assess whether they were answered correctly or require new corrections. I request that the exact location of the changes marked at each point, page, and line be inserted.

Round 3

Reviewer 4 Report

Comments and Suggestions for Authors

Dear authors,

 Thanks for your reply. This new version is better. All questions were clarified and I don't have any asks. 

Congrats.

Author Response

 Thanks for your reply. This new version is better. All questions were clarified and I don't have any asks. 

Congrats.

Answer: Thank you very much for reviewing our manuscript.